# Comparing effects of wearable robot-assisted gait training on functional changes and neuroplasticity: A preliminary study

Jungsoo Lee[1]*, Kassymzhomart Kunanbayev[2], Donggon Jang[2], Dae-Shik Kim[2]

1 Department of Medical IT Convergence Engineering, Kumoh National Institute of Technology, Gumi, Republic of Korea, 2 School of Electrical Engineering, Korea Advanced Institute of Science and Technology, Daejeon, Republic of Korea

* jungsoo0319@gmail.com

**Data Availability Statement:** All relevant data are within the manuscript and its Supporting Information files.

**Funding:** This study was supported by T-ROBOTICS (G01210547) and the National

## Abstract

Robot-assisted gait training (RAGT) is a promising technique for improving the gait ability of elderly adults and patients with gait disorders by enabling high-intensive and task-specific training. Gait functions involve multiple brain regions and networks. Therefore, RAGT is expected to affect not just gait performance but also neuroplasticity and cognitive ability. The purpose of this preliminary study was to verify the feasibility of the proposed RAGT design and to assess and compare the effect sizes of various measurement variables, including physical, cognitive, and neuroimaging induced by RAGT. Twelve healthy adults without any neurological or musculoskeletal disorders participated in this study. All participants wore a wearable exoskeleton robot and underwent 10 RAGT sessions. Functional data related to physical and cognitive abilities and neuroimaging data obtained from a magnetic resonance imaging (MRI) scanner and a functional near-infrared spectroscopy (fNIRS) device were acquired before and after the training sessions to assess the effect sizes of variables affected by RAGT. All participants underwent 10 sessions of RAGT without any adverse incidents, and the feasibility of the proposed RAGT design, consisting of preferred speed walking, fast speed walking, inclined walking, and squats, was validated. Variables related to physical and cognitive abilities significantly improved, but those related to neuroplasticity did not. The effect size of physical ability was "very large," whereas that of cognitive ability was "medium-to-large." The effect sizes of functional and structural neuroplasticity showed "medium" and "very small," respectively. The effect size of the RAGT varied depending on the measured variables, with the effect size being the greatest for physical ability, followed by cognitive ability, functional neuroplasticity, and structural neuroplasticity. The proposed RAGT design affects cognitive and neuroplastic effects beyond the physical effect directly affected by RAGT. This study highlights that while RAGT can positively influence cognitive outcomes beyond physical benefits, more intensive interventions may be required to elicit significant neuroplastic changes. This preliminary study offers useful information for researchers interested in designing robot-assisted training by investigating the potential extent of neuroplastic effects.

**Trial registration:** KCT0006738.

Research Foundation of Korea (NRF) grant funded by the Korean government (MSIT). (No. RS-2023-00208884).

**Competing interests:** The authors have declared that no competing interests exist.

## Introduction

Gait function is important because it is related to mobility, activities of daily living, and quality of life [1, 2]. Recently, robot-assisted gait training (RAGT), which uses advanced electromechanical devices to support body weight and automate the gait process, has been used to enhance the gait function of people whose mobility has been affected by aging and neurological disorders [3–7]. RAGT can provide highly intensive and repetitive training with consistent and precise movements, which can be difficult to achieve with conventional therapy [8]. Because these characteristics of RAGT are crucial for motor learning and neuroplasticity, RAGT is particularly beneficial for individuals with aging and neurological impairment. According to previous clinical studies, RAGT improved gait function, including gait speed, cadence, stride length, step width, single support time, and cardiopulmonary metabolic efficiency in elderly adults [3]. Also, in RAGT studies for patients with neurological disorders, gait speed, endurance, balance, and functional ambulation were improved in stroke, Parkinson's disease, and spinal cord injury patients, and abnormal gait patterns and muscle strength were better in children with cerebral palsy [4–7].

Gait is a representative function of the musculoskeletal system and is also associated with the central nervous system, including multiple cognitive domains [9, 10]. Previous gait studies reported that slow gait speed is related to cognitive decline, including worse executive function, attention, and memory [11–13]. Moreover, slow gait speed is a key symptom of several neurodegenerative diseases such as Alzheimer's disease and Parkinson's disease [14]. In terms of neurophysiological mechanisms, gait is associated with complex processes regulated by networks of subcortical and cortical various regions in the brain beyond the sensorimotor network [15]. Important regions of gait control include sensorimotor cortical areas (primary motor cortex, supplementary motor area, premotor cortex, somatosensory cortex), prefrontal cortical areas, basal ganglia, thalamus, and cerebellum [15]. The initiation of voluntary movement originates in the prefrontal cortex, where it generates signals that are transmitted to the motor cortical areas to initiate a motor response. Signals from the motor cortical areas are then sent down to two basal ganglia-thalamus and cerebellum-thalamus loops [16]. Gait function involves multiple regions and networks of the central nervous system, as well as the musculoskeletal system. Accordingly, RAGT is anticipated to influence not only gait performance but also cognitive function and neuroplasticity. Several studies reported that RAGT improved cognitive function, such as dual-task performance and cognitive assessment score, in stroke patients [17–19]. Also, previous neuroimaging studies reported that RAGT modulated brain activation and power in the specific frequency band in the motor-specific areas and facilitated white matter integrity in stroke patients [20–22]. Previous studies have separately examined the effects of RAGT on cognitive function, functional neuroplasticity, and structural neuroplasticity, respectively. To the best of our knowledge, studies that simultaneously investigate cognitive function, functional neuroplasticity, and structural neuroplasticity within a single study are limited.

This study explored the changes in physical and cognitive abilities caused by RAGT and the effects of gait and cognition-related functional and structural neuroplasticity using magnetic resonance imaging (MRI) and functional near-infrared spectroscopy (fNIRS) data before and after RAGT in healthy adults. This study provides effect seizes of physical function, cognitive function, functional neuroplasticity, and structural neuroplasticity induced by RAGT and preliminary insights into potential neuroplastic effects associated with RAGT to guide future studies. In addition, it offers useful information for researchers interested in designing robot-assisted training by investigating the potential extent of neuroplastic effects.

## Materials and methods

### Ethics statement

This study was approved by Institutional Review Board of the Korea Advanced Institute of Science and Technology (KH2021-180) on October 8, 2021. The researcher read and explained the consent form to the participants while reviewing the document together and answered any questions they had. Finally, the researcher confirmed the participants' understanding of the research content and the use of personal information and obtained written consent for participation in the study. All participants voluntarily agreed to participate.

### Participants and experimental design

In total, 12 healthy adults (10 males, 2 females; mean age: 25.5 ± 3.7 years) were enrolled in this study. All participants were recruited from the community between February 3, 2022 and August 18, 2022. The inclusion criteria were as follows: 1) age 19–64 years, and 2) no history of central nervous system diseases, lower extremity musculoskeletal disorders, or cardiopulmonary diseases. Participants were excluded if they 1) had difficulty understanding and continuously participating in the research, 2) had a body that was not suitable for the size of the wearable walking assistance robot, or 3) had any contraindications to MRI.

All participants visited twice weekly and received 10 sessions of robot-assisted training over 5 consecutive weeks. Each training session involved wearing a wearable robot (Myosuit, MyoSwiss AG, Zurich, Switzerland) and performing the following tasks: walking at a preferred speed on a treadmill for warm-up, fast speed walking on a treadmill, inclined walking on a treadmill, and squatting on the ground (Fig 1A).

The Myosuit, a fully untethered and autonomous soft, cable-driven exosuit weighing 5.5 kg, is designed to assist across daily life activities and to enable intensive training for strength, endurance, and balance activities with minimal interference with the user's movements [23]. The exosuit in use in shown in Fig 1. Therefore, the Myosuit is suitable as this study device for exploring the training effects of high-intensity lower limb exercises in daily activities for healthy individuals. The device consists of a backpack-style motor unit, a textile vest with a waist belt, and polymer knee orthoses. Adjustable polymer springs assist passive hip flexion,

**A. Training Design**

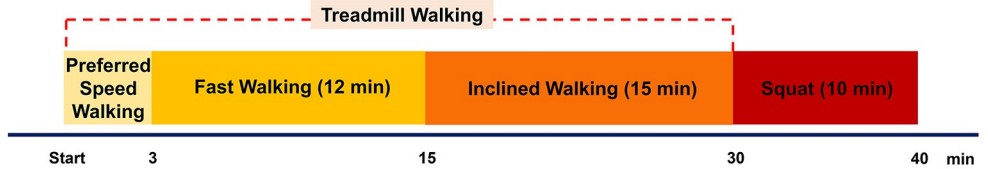

**B. Dual Task Walking Design for fNIRS Data Acquisition**

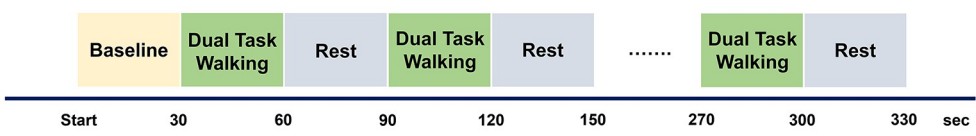

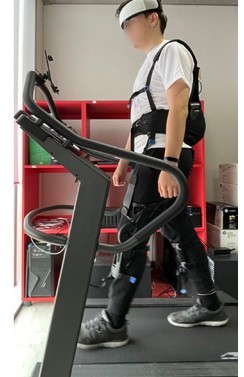

**Fig 1. Study design and a participant with the Myosuit.** (A) Robot-assisted gait training design. All participants visited twice weekly and received 10 sessions of robot-assisted training. (B) Dual task walking design. The phonemic verbal fluency task was required during gait on the treadmill before and after 10 training sessions. This dual task was performed twice with the robot and without the robot, and the condition order was randomly determined.

while ultra-high-molecular-weight polyethylene cables, driven by electric motors in the backpack, provide active assistance. The cables are routed across the user's lower body, coupling hip and knee extension in an underactuated manner. The Myosuit can automatically detect gait phases and assist the knee and hip extensor muscles using one cable routed across the hip and knee joints and passive elastic bands. During walking, cable tension is applied during the stance phase to assist hip and knee extension. The assistance level can be customized for each leg, providing a high level of personalization to meet the user's training requirements [23].

For all training sessions, treadmill speed and slope, slope change time, and number of squats were recorded to evaluate improvements in physical function. To investigate changes in cognitive function, neural activity, and plasticity, fNIRS data and the number of correct answers were acquired during the cognitive dual task, and MRI data were acquired during rest. All measurements were taken before and after 10 training sessions.

## Robot-assisted training

RAGT generally involves gait training either on a treadmill or overground at the preferred waking speed or fast walking speed. Additionally, activities such as up-hill and down-hill walking, stairs ascending and descending, and squatting can be selected based on the participant's functional status or training goal [24, 25]. In this study, RAGT involved the preferred speed walking, fast speed walking, inclined walking on a treadmill, and squatting on the ground for 40 minutes. Thus, the training was composed of a combination of conventional exercises. The unique aspect lies in the personalization of treadmill incline during inclined walking, adjusted according to the individual's physical capacity and training effects. For all training sessions, the "Assist" mode was used for assistance with knee and hip extensions. The treadmill slope was set at 0% for the preferred and fast speed walking. The preferred walking speed was set according to the natural walking speed at which individual participants felt comfortable. The fast speed walking was set at 150% of the preferred walking speed. According to previous studies [26–28], fast walking speeds range between 120–150% of the preferred walking speed. In this study, as it involves relatively high-intensive training, the speed was set at 150%. For inclined walking, the treadmill slope was initially set at 35%, and the speed was set to the preferred walking speed. The treadmill slope started at 35%, and when the participants expressed exhaustion, the slope was reduced by 5% to complete the 15 minute training. The participants' physical strength increased as the training was repeated. After obtaining participant consent, we increased the starting slope by 5% to enable intensive training. Squatting was performed on the ground. To help participants maintain the correct posture and a constant pace, the experimenter faced the participants and conducted the training with them. Squats were performed 10 times per set, the break time between sets was 10 s, and the break time between sets was extended at the request of the participants.

## MRI data acquisition and processing

MRI data, including functional MRI (fMRI), diffusion tensor imaging (DTI), and 3D T1-weighted imaging, were acquired using a Siemens Magnetom Verio 3T MR scanner (Siemens Healthcare, Erlangen, Germany). The participants were instructed to keep their eyes closed, not think about anything in particular, and remain motionless during scanning. The imaging data were acquired with the following settings: fMRI data, 300 volumes, 36 axial slices, slice thickness = 3 mm, matrix size = $64 \times 64$, repetition time = 2000 ms, echo time = 30 ms, and field of view = $192 \times 192$ mm; DTI data, b = 1000 s/mm$^2$, 64 non-colinear gradient directions, 48 axial slices, slice thickness = 2.5 mm, matrix size = $256 \times 256$, repetition time = 9700 ms, echo time = 93 ms, and field of view = $230 \times 230$ mm; T1-weighted structural data, 256

axial slices, slice thickness = 1 mm, matrix size = 176 × 256, repetition time = 1800 ms, echo time = 2.52 ms, flip angle = 9˚, and field of view = 263 × 350 mm.

To construct functional networks and extract network measures, fMRI data were preprocessed using the CONN toolbox version 21a (McGovern Institute for Brain Research, Massachusetts Institute of Technology, Cambridge, MA, USA; http://www.nitrc.org/projects/conn) implemented in MATLAB. Head motion, slice-timing corrections, and outlier detection for scrubbing were performed. Additionally, registration, segmentation, and spatial smoothing with an 8 mm full-width-at-half-maximum Gaussian kernel were performed. Nuisance signals were removed by linear regression. The confounding factors in the regression included six head motion parameters and six first-order temporal derivatives of the motion parameters, each of the five parameters obtained from the principal component analysis of the temporal components of the white matter and ventricle signals, and scrubbing parameters for outlier volumes. Bandpass filtering (0.008–0.09 Hz) and linear detrending were performed.

The regions in the functional network were determined using thirty-two Montreal Neurological Institute (MNI) coordinates of peak activation during leg movements in a previous study [29]. The network edge, which indicates the connection strength, was constructed by calculating Pearson's correlation coefficients between the mean time courses within each region, defined as the 6 mm diameter sphere around each MNI coordinate. The regions of interest and MNI coordinates are listed in S1 Table in S1 File.

Network efficiency, which is a graph-theoretical measure, was used to extract the network topological measure. This metric measures how efficiently information is exchanged in a network by calculating the inversion of the minimum number of connections required to travel from one network node to another and is defined as follows [30–32]:

$$Network\ efficiency = \frac{1}{n}\sum_{i \in N}\frac{\sum_{j \in N, j \neq i}(d_{ij}^{w})^{-1}}{n-1}$$

where $n$ is the number of regions, and $d_{ij}^{w}$ is the shortest path length between regions $i$ and $j$.

To extract the white matter integrity of the vertical, horizontal, and lateral major nerve tracts, such as the corticospinal tract (CST), corona radiata (CR), superior longitudinal fasciculus (SLF), and corpus callosum (CC), DTI data were preprocessed using the FDT (FMRIB's Diffusion Toolbox) implemented in the FSL software package 5.0.9 (FMRIB Software Library, FMRIB, Oxford, UK, http://www.fmrib.ox.ac.uk/fsl). Corrections for eddy currents, head motion, and skull stripping were performed. The *DTIfit* algorithm, which is most often used to quantify white matter integrity, was used to fit a tensor model and reconstruct a functional anisotropy (FA) map. Spatially normalized FA maps were reconstructed by registering individual FA maps to the MNI standard space (FMRIB58_FA standard space image) using nonlinear registration algorithms of the tract-based spatial statistics (*TBSS*) technique.

The CST template descending from the primary motor cortex was obtained from a previous probabilistic tractography study [33], and the CR, SLF, and CC obtained from the Johns Hopkins University White Matter Atlas (JHU ICBM-DTI-81) [34] were used to determine white matter integrity. All tracts were binarized and masked on spatially normalized FA maps. The FA values were obtained by averaging within the regions in the left and right hemispheres.

## fNIRS data acquisition and processing

The fNIRS data were acquired using a portable 48-channel device (NIRSIT, OBELAB Inc., Seoul, Korea) with wavelengths of 780 and 850 nm based on the modified Beer-Lambert Law. The source-detector pairs of the device attached to the forehead measured the oxy-hemoglobin (HbO) concentration in 48 regions of the prefrontal cortex. HbO concentration data of the

prefrontal cortex were acquired during the cognitive dual task on a treadmill before and after 10 training sessions (Fig 1B). The participants were asked to perform a phonemic verbal fluency task while walking on a treadmill. The participants in the phonemic verbal fluency task were required to listen to the consonant presented by the experimenter and generate as many words as possible that started with the given consonant within a 15 second timeframe (two consonants per dual task walking block) [35]. The number of correct and non-repetitive words generated by the participant that begin with the given consonant during the dual task time-frame is counted and recorded by the experimenter. This dual task was performed twice with and without the robot, and the condition order was randomly determined.

The collected fNIRS data were preprocessed using the NIRSIT Analysis Tool (OBELAB Inc., Seoul, Korea). The raw data was low- and high-pass filtered at 0.1 Hz and 0.05 Hz with a discrete cosine transform. Channels with more than five undefined values, five negative frames, and a signal-to-noise ratio (SNR) below 30 dB were rejected (SNR was calculated based on a time series between 10 and 15 s). Furthermore, for each subject, outlier channels were removed based on a visual inspection of the hemoglobin concentration. A general linear model analysis of the preprocessed data was conducted using the HbO concentration time series of all 48 channels. A motion regressor with acceleration was utilized with a filter (low-pass at 0.1 Hz, high-pass at 0.005 Hz) applied to the inertial measurement unit (IMU) embedded in the fNIRS device.

## Statistical analysis

The Shapiro-Wilk normality test was used to evaluate data normality. The null hypothesis was rejected in the data values. A paired t-test was conducted to evaluate significant changes in physical and cognitive functional variables and neuroimaging variables after 10 training sessions. Cohen's d was used to evaluate the effect size of changes in these variables. Cohen's d = 0.01, 0.20, 0.50, 0.80, and 1.30 are interpreted as very small, small, medium, large, and very large effects, respectively [36, 37]. Two-sample t-testing was conducted to evaluate significant differences in a cognitive functional variable and an fNIRS variable obtained from cognitive dual task between with and without wearable robot conditions. These statistics were derived using the statistics toolbox of MATLAB R2022a (Mathworks, Natick, MA, USA), and the fNIRS data were derived using the NIRSIT Analysis Tool (OBELAB Inc., Seoul, Korea).

## Results

All 12 participants completed 10 sessions of robot-assisted training over five consecutive weeks without any adverse events. The acceptability and feasibility of Myosuit-based RAGT were validated. In the first training session, the average speed of the treadmill for preferred speed walking was 4.06±0.09 km/h, and the average speed of the treadmill for fast speed walking was 6.08±0.14 km/h. Improvement in physical ability due to robot-assisted training was examined through changes in the treadmill slope during inclined walking and the number of sets during squats (Fig 2). The initial average slope of the treadmill for inclined walking was 30.29±3.58%, and the initial average set of squats was 12.08±1.93. In the final training session, the average slope of the treadmill for inclined walking was 34.39±3.79%, and the average set of squats was 13.67±1.83. The treadmill slope and number of squats significantly improved through robot-assisted training (treadmill slope, t = 4.56, p<0.001; number of squats, t = 2.37, p = 0372). The changes in physical ability showed a "very large" effect size in inclined walking and a "medium" effect size in squats (treadmill slope, d = 1.32; number of squats, d = 0.68).

The improvement in cognitive ability during gait was examined through changes in the number of correct answers during the cognitive dual task before and after 10 training sessions.

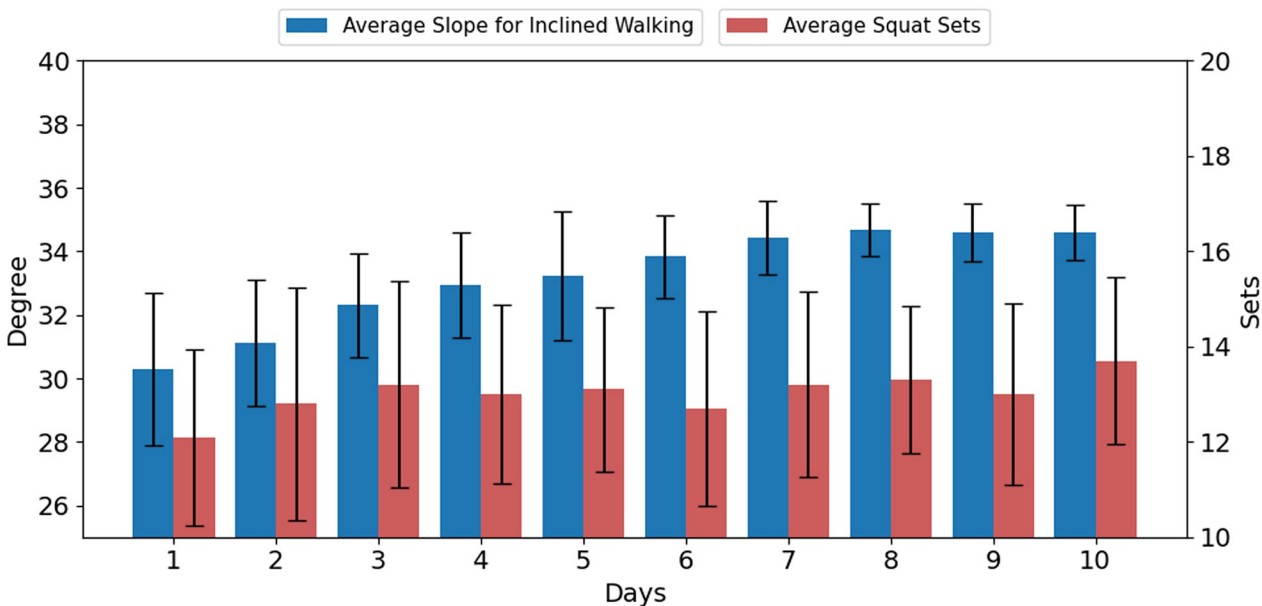

**Fig 2. Changes in the average slope for inclined walking and average squat sets over 10 robot-assisted gait training sessions.**

The number of correct answers refers to the count of accurate and non-repetitive words generated by the participant during the verbal fluency task while walking on a treadmill. The numbers of correct answers during the dual task were 45.42±11.16 (with robot) and 46.08±8.40 (without robot) at pre-training. No significant differences were observed among the initial conditions (t = 0.17, p = 0.8702). At post-training, the number of correct answers was 50.67 ±10.33 (with robot) and 53.17±11.45 (without robot). The number of correct answers in the phonemic verbal fluency task significantly improved through robot-assisted training (with robot, t = 2.47, p = 0.0311; without robot, t = 4.32, p = 0.0012). In the changes in correct answers during the cognitive dual task with robot, a medium effect size was observed (d = 0.71), while a large effect size was observed in the dual task without the robot (d = 1.25), indicating an overall medium-to-large effect size.

The effects of functional and structural neuroplasticity were examined by investigating changes in the functional network and white matter integrity of the major nerve tracts. An overall trend of increase in functional network connectivity strength was observed (Fig 3A). Network efficiency had no statistical significance but tended to increase after 10 training sessions (t = 1.88, p = 0.0872) (Fig 3B). The effect size of functional neuroplasticity was "medium" (d = 0.54). The mean values of white matter integrity of the major nerve tracts, including the CST, CR, CC, and SLF, slightly increased, but the difference was not statistically significant. The effect size of structural neuroplasticity was "very small" (CST, d = 0.02; CR, d = 0.12; CC, d = 0.05; SLF, d = 0.11). The changes in physical, cognitive, and neuroplastic measures are summarized in Table 1. The effect size for physical ability was categorized as "very large," while cognitive ability demonstrated a "medium-to-large" effect size. The effect sizes for functional and structural neuroplasticity were "medium" and "very small," respectively. The effect of RAGT varied across the different measures, showing the largest effect size for physical ability, followed by cognitive ability, functional neuroplasticity, and structural neuroplasticity. The data of physical, cognitive, and neuroplastic measures are detailed in S1 File.

Cognition-related activity during gait was investigated using fNIRS measurements during the dual task with and without the robot (Fig 4). Increased activation was observed in the

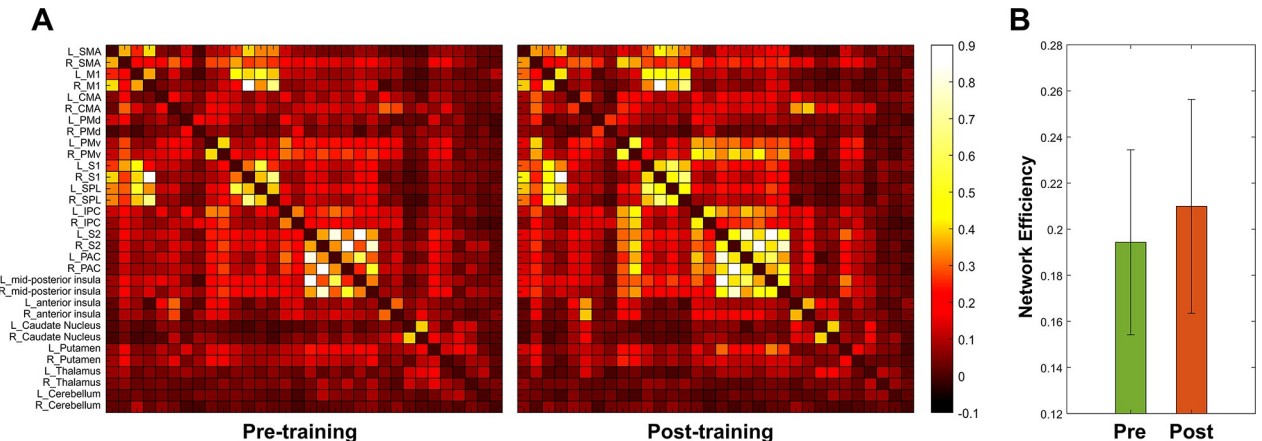

**Fig 3. Altered gait-related functional network obtained from resting-state fMRI data.** (A) Functional network at pre-and post-training. The color bar indicates connectivity strength. (B) Network efficiency of the functional network at pre-and post-training. L, left; R, right; SMA, supplementary motor area; M1, primary motor cortex; CMA, cingulate motor area; PMd, dorsal premotor cortex; PMv, ventral premotor cortex; S1, primary somatosensory area; SPL, superior parietal lobule; IPC, inferior parietal cortex; S2, secondary somatosensory area; PAC, primary auditory cortex.

prefrontal cortex during the dual tasks. The dual task with the robot tended to have lower pre-frontal activity than that without the robot before and after training. After 10 training sessions, most of the prefrontal cortex activity during the dual task showed a decreasing pattern compared with pre-training except for the right ventrolateral prefrontal and orbitofrontal cortical regions in the dual task without the robot.

## Discussion

This preliminary study investigated changes in physical and cognitive abilities and the effects of RAGT on gait and cognition-related neuroplasticity in healthy adults. The results suggested that the effect size varied depending on the measured variables; however, a positive effect was generally observed.

**Table 1. Changes in physical, cognitive, and neuroplastic measures.**

| Measures | Pre | Post | *p* | Cohen's d | ES |
|---|---|---|---|---|---|
| **Physical measures** | | | | | |
| Treadmill slope for inclined walking (%) | 30.29 (3.58) | 34.39 (3.79) | <0.001 | 1.32 | Very large |
| Set of squats (n) | 12.08 (1.93) | 13.67 (1.83) | 0.0372 | 0.68 | Medium |
| **Cognitive measures** | | | | | |
| Correct answers for cognitive dual task with robot (n) | 45.42 (11.16) | 50.67 (10.33) | 0.0311 | 0.71 | Medium |
| Correct answers for cognitive dual task without robot (n) | 46.08 (8.40) | 53.17 (11.45) | 0.0012 | 1.25 | Large |
| **Neuroplastic measures** | | | | | |
| Network efficiency | 0.1943 (0.0401) | 0.2099 (0.0464) | 0.0872 | 0.54 | Medium |
| Tract integrity of CST | 0.5821 (0.0267) | 0.5827 (0.0323) | 0.9503 | 0.02 | Very small |
| Tract integrity of CR | 0.4738 (0.0188) | 0.4756 (0.0221) | 0.6790 | 0.12 | Very small |
| Tract integrity of CC | 0.6498 (0.0173) | 0.6510 (0.0243) | 0.8554 | 0.05 | Very small |
| Tract integrity of SLF | 0.4562 (0.0261) | 0.4589 (0.0283) | 0.7013 | 0.11 | Very small |

Data are expressed as the mean (standard deviation). ES, effect size; CST, corticospinal tract; CR, corona radiata; CC corpus callosum; SLF, superior longitudinal fasciculus.

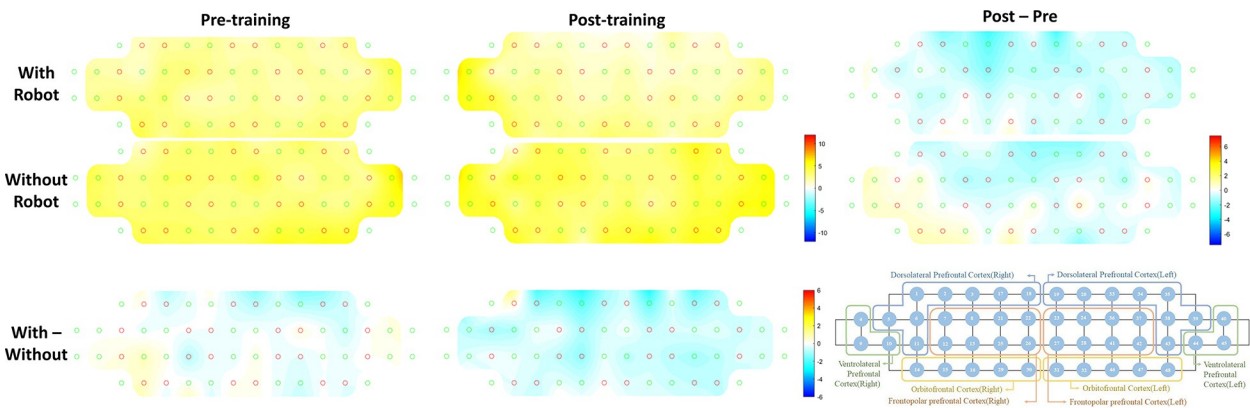

**Fig 4. Average activation of the prefrontal cortex during cognitive dual task obtained from fNIRS data of all trials and participants.** The color bar indicates the t-value. Yellow-red represents increased activity, and cyan-blue represents decreased activity.

In terms of physical ability, the variables significantly increased after 10 training sessions. Evidence related to the improvement of physical ability by RAGT has been reported in older adults and patients with neurological diseases [38–40]. Previous studies have reported that RAGT enhances muscle strength and endurance, improves gait speed and balance ability, and decreases gait metabolic demand [38–40]. These results are consistent with the fact that RAGT acts directly on the lower extremities and supports gait patterns. In this study, the change in the treadmill slope during inclined walking, which reflects muscle strength and endurance, showed a "very large" effect size. The effect size of the changes in squat sets performed after inclined walking was "medium." The reason for the effect size of the squat being lower than that of inclined walking is attributed to the training performed after the inclined walking sessions, where the slope angle was progressively increased by the session, leading to participant exhaustion.

Cognitive ability during gait was assessed using the cognitive dual task of performing a phonemic verbal fluency task while continuing to walk on the treadmill. The number of correct answers for the phonemic verbal fluency task before and after the 10 training sessions significantly improved and showed a "medium" to "large" effect size. The "medium" effect size was shown in the dual task with the robot condition, and the "large" effect size was shown in the dual task without the robot condition. Previous studies have reported that physical exercise can improve cognitive functions and have discovered epigenetic changes caused by physical exercise [41, 42]. In particular, gait control regions involve the prefrontal cortical areas, basal ganglia, thalamus, and cerebellum, as well as motor-related regions [15]. Moreover, gait studies have reported that slow gait speed is related to cognitive decline, including worse executive function, attention, and memory [11–13]. These previous results are in line with our study results that RAGT helps improve cognitive ability. In this study, the effect size was smaller than that of physical ability, indicating the "very large" effect size. RAGT is a "bottom-up" intervention that affects the brain by directly affecting the lower extremities using complete assistance or assistance-as-needed resistance [43]. Therefore, it can be assumed that the effect of the cognitive ability was lower than the effect of physical ability directly associated with RAGT.

The effects of RAGT on functional and structural neuroplasticity were investigated using neuroimaging data. Although no statistically significant results or relatively small effect sizes were observed in our study, a positive trend of RAGT on brain plasticity was identified. The effect size of functional neuroplasticity was "medium" and that of structural neuroplasticity

was "very small." Given that our study participants were healthy adults, it may be difficult to observe structural brain changes within the current study design. However, the effect on functional neuroplasticity is noteworthy. The overall trend of an increase in gait-related functional network connectivity strength was noticeable. Network efficiency, which measures how efficiently information is exchanged in a network, showed a marginally significant change. Although smaller than the effect sizes of physical and cognitive abilities, RAGT seems to be able to increase the efficiency of gait-related networks. In addition, after 10 training sessions, most prefrontal cortex activity during the cognitive dual task showed a decreasing pattern compared to pre-training. Due to the RAGT, it can be interpreted that less prefrontal cortex function is utilized during cognitive activities while walking. This result is meaningful because it is a brain activity pattern that has been shown in situations where the number of correct answers in a phonemic verbal fluency task during walking is significantly improved. These neuroimaging results suggest that RAGT could potentially contribute to improved gait-related network efficiency and more effective use of cognitive resources during walking. A higher fall risk was observed in cognitively impaired older adults; therefore, a lack of cognitive resources during gait is considered a major factor in falls [44]. In this regard, RAGT has the potential to help prevent falls. The effect size of neuroplasticity was the smallest among the measured variables. Despite changes in physical and cognitive abilities, a more intensive RAGT intervention may be required to induce changes in the brain. The dual task with the robot condition showed lower prefrontal activity than the dual task without the robot condition pre-and post-training. Wearable robots that can be comfortably worn during daily activities may assist in the efficient use of cognitive resources during walking.

Our findings provide evidence that RAGT can contribute to the improvement of physical and cognitive abilities and modulate functional neuroplasticity. The effect size of the RAGT varied depending on the measured variables, with the effect size being the greatest for physical ability, followed by cognitive ability, functional neuroplasticity, and structural neuroplasticity. This study had several limitations. First, this was a single-group study with a small sample size. Moreover, this preliminary study aimed only to evaluate the acceptability and feasibility of Myosuit-based RAGT and measure the effect size of physical, cognitive, and neuroimaging variables within the current intervention design. Second, there were not enough measured variables to examine the changes in physical and cognitive abilities in detail. Despite the availability of various cognitive function assessment methods, the use of the number of correct answers for the cognitive dual task was insufficient to measure the participants' cognitive ability. However, measuring cognitive function while walking in relation to falls, which are highly related to elderly mortality, is meaningful and will be useful in explaining the fall prevention effect of RAGT if it is developed and utilized in a better design. Based on the data from this preliminary study, we will decide on the sample size and plan a parallel-arm design with a larger sample size and more variables.

This preliminary study confirmed the feasibility of implementing a high-intensity RAGT training protocol utilizing the device, as all participants successfully completed the proposed training program. Additionally, we assessed the effect sizes induced by the RAGT from various perspectives, including physical function, cognitive function, and neuroplasticity, by measuring physical, cognitive, functional and structural neuroimaging measures before and after the training. The effect sizes were observed in the following order: physical function, cognitive function, and neuroplastic changes. RAGT demonstrated significant effects on both physical and cognitive functions in this study. From a neuroplasticity perspective, while a positive trend was noted, the effect was relatively weak. This finding suggests that more intensive interventions and personalized strategies in terms of intervention design may be necessary to induce neuroplastic changes beyond the improvements observed in physical and cognitive

functions. Furthermore, the measurement methods for physical, cognitive, and neuroplastic changes, as well as the quantitative information regarding the effect sizes of neuroplastic changes compared to those of physical and cognitive functions, may assist in designing future studies, including intervention, examination, and sample size calculation, for researchers interested in the neuroplastic effects induced by RAGT.

## Supporting information

**S1 File. Study dataset.**
(PDF)

## Acknowledgments

We would like to thank all participants who consented to participate in the study.

## Author Contributions

**Conceptualization:** Jungsoo Lee, Dae-Shik Kim.

**Data curation:** Jungsoo Lee, Kassymzhomart Kunanbayev, Donggon Jang.

**Formal analysis:** Jungsoo Lee, Kassymzhomart Kunanbayev, Donggon Jang.

**Funding acquisition:** Dae-Shik Kim.

**Investigation:** Jungsoo Lee, Kassymzhomart Kunanbayev, Donggon Jang, Dae-Shik Kim.

**Methodology:** Jungsoo Lee.

**Project administration:** Jungsoo Lee, Dae-Shik Kim.

**Resources:** Jungsoo Lee.

**Supervision:** Dae-Shik Kim.

**Validation:** Jungsoo Lee.

**Writing – original draft:** Jungsoo Lee.

**Writing – review & editing:** Jungsoo Lee, Dae-Shik Kim.

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
