## [Decision Letter · Decision Letter 0]

3 Sep 2024

PONE-D-24-20412Comparing Effects of Wearable Robot-assisted Gait Training on Functional Changes and Neuroplasticity: A Preliminary StudyPLOS ONE

Dear Dr. Lee,

Thank you for submitting your manuscript to PLOS ONE. After careful consideration, we feel that it has merit but does not fully meet PLOS ONE’s publication criteria as it currently stands. Therefore, we invite you to submit a revised version of the manuscript that addresses the points raised during the review process.

We look forward to receiving your revised manuscript.

Kind regards,

Noman Naseer, PhD

Academic Editor

PLOS ONE

Journal Requirements:

Additional Editor Comments:

I have recieved comment from two reviewers and both have suggested some major revisions. Please revise upon those comment and submit again.

Reviewers' comments:

Reviewer's Responses to Questions

**Comments to the Author**

1. Is the manuscript technically sound, and do the data support the conclusions?

Reviewer #1: No

Reviewer #2: Yes

2. Has the statistical analysis been performed appropriately and rigorously? 

Reviewer #1: No

Reviewer #2: Yes

3. Have the authors made all data underlying the findings in their manuscript fully available?

Reviewer #1: No

Reviewer #2: Yes

4. Is the manuscript presented in an intelligible fashion and written in standard English?

Reviewer #1: Yes

Reviewer #2: Yes

5. Review Comments to the Author

Reviewer #1: In this paper, the technique of Robot-Assisted Gait Training (RAGT) is established to show improvement in gait training for patients with gait disorders. The technique is insightful and offers interesting outcomes. However, there are certain points which I would like the authors to consider amending in this document as follows:

1. The abstract does not explicitly link the findings with the actual objective of the study, set out in the beginning of the abstract. Please revise the latter sections of the abstract.

2. In the introduction section, the authors need to ground the study in the previous literature, which has not been done.

3. Moreover, the introduction section should elaborate about the RAGT technique with evidence and justification.

4. The methodology section lacks addressal to the exact model and type of wearable RAGT device and training were utilized in the experiment. Moreover, there is no pictorial evidence and no citation-based evidence to justify the use of the particular Myosuit.

5. In the process of RAGT, there needs to be justification of why 150% of the preferred walking speed was kept.

6. Citations for how previous similar trainings were done are needed in the methodology section.

7. The method for gauging accuracy of answers in the tasks is unclear.

8. Further explanation is needed for explaining medium to large effect size in the findings section.

9. A control and experimental group were needed to measure the efficacy of the RAGT.

10. Statistical analysis and verification are needed to gauge the ‘positive effect of RAGT on brain plasticity’.

11. The findings section is very sketchy and needs to include vital comparisons and analysis to establish significance of the outcome.

12. The manuscript lacks a section on conclusion, which is very important to comprehensively summarize the whole study.

Reviewer #2: The manuscript presents the feasibility of the robot assisted gait-training (RAGT)-based study design to evaluate the physical, cognitive, and neuroimaging effect. 10 healthy adults participated in this study. Walking (preferred sped, fast, and squats) with and without RAGT on treadmill with and without inclined surface is practiced to measure physical strength, magnetic resonance imaging (MRI) scanner and a functional near-infrared spectroscopy (fNIRS) device is used to measure and evaluate cognitive and neuroimaging (cognitive and neuroplasticity) behavior. The manuscript is well written and well structured.

Following are the suggestions to improve the quality of manuscript.

1. The sample size is very small to estimate the results or impact of the study. Authors mentioned total of 12 healthy adults out of which 10 are male adults but no information is presented regarding the rest of two participants.

2. The discussion section only includes few studies to verify and compare results. All the studies verified the current study results but this is very limited in scope of this study. Although this is mentioned in the limitations, however this limited the novelty and contribution of this study. If all the conclusions are already explored in the literature then what this study implies?

3. In cognitive-related activity, why all the channels are placed at prefrontal cortex, since the participant is performing motor activity, premotor cortex should be included it will provide additional insights of combined mental workload and cognitive-related activity both in prefrontal and premotor cortex.

4. Does Fig-4 represents average cognitive activity of all trials and participants or single-trial of example participant? This should be mentioned in the caption or description.

5. The pixel quality of fig-3 is not good and it is difficult to analyze the figure. Please improve the quality of this figure.

6. There are many mistakes in references formatting, please review thoroughly.

7. The contribution of the study with respect to neuroplastic effect are very limited. The authors discussed the limitation in the discussion section “Although no statistically significant results or relatively small effect sizes were observed in our study” mainly because of small sample size. This decrease the impact, novelty and contribution of this study as compared to the claims of this study.

6. PLOS authors have the option to publish the peer review history of their article (what does this mean?). If published, this will include your full peer review and any attached files.

Reviewer #1: No

Reviewer #2: No

---

## [Author Response · Author response to Decision Letter 0]

16 Oct 2024

We sincerely appreciate the editor and reviewers for their valuable feedback. We have carefully addressed their concerns and incorporated the suggested changes into the revised manuscript and materials. Please refer to the "Response to Reviewers" file for more information.

---

## [Decision Letter · Decision Letter 1]

21 Nov 2024

Comparing effects of wearable robot-assisted gait training on functional changes and neuroplasticity: a preliminary study

PONE-D-24-20412R1

Dear Dr. Lee,

We’re pleased to inform you that your manuscript has been judged scientifically suitable for publication and will be formally accepted for publication once it meets all outstanding technical requirements.

Kind regards,

Noman Naseer, PhD

Academic Editor

PLOS ONE

Additional Editor Comments (optional):

The paper has been revised well.

Reviewers' comments:

Reviewer's Responses to Questions

**Comments to the Author**

1. If the authors have adequately addressed your comments raised in a previous round of review and you feel that this manuscript is now acceptable for publication, you may indicate that here to bypass the “Comments to the Author” section, enter your conflict of interest statement in the “Confidential to Editor” section, and submit your "Accept" recommendation.

Reviewer #1: All comments have been addressed

Reviewer #2: All comments have been addressed

2. Is the manuscript technically sound, and do the data support the conclusions?

Reviewer #1: Yes

Reviewer #2: Yes

3. Has the statistical analysis been performed appropriately and rigorously? 

Reviewer #1: Yes

Reviewer #2: Yes

4. Have the authors made all data underlying the findings in their manuscript fully available?

Reviewer #1: No

Reviewer #2: Yes

5. Is the manuscript presented in an intelligible fashion and written in standard English?

Reviewer #1: Yes

Reviewer #2: Yes

6. Review Comments to the Author

Reviewer #1: I have thoroughly checked the revisions, done on the basis of my previous comments, and they seem satisfactory. The authors have worked on explaining the RAGT technique, providing information on statistical analysis, and establishing the need for it. Their incorporations and justifications deem appropriate now.

Reviewer #2: (No Response)

7. PLOS authors have the option to publish the peer review history of their article (what does this mean?). If published, this will include your full peer review and any attached files.

Reviewer #1: No

Reviewer #2: **Yes: **Hammad Nazeer

---

## [Editor Report · Acceptance letter]

25 Nov 2024

PONE-D-24-20412R1 

PLOS ONE

Dear Dr. Lee, 

I'm pleased to inform you that your manuscript has been deemed suitable for publication in PLOS ONE. Congratulations! Your manuscript is now being handed over to our production team.

Kind regards, 

on behalf of

Dr. Noman Naseer 

Academic Editor

PLOS ONE